# Early correction of synaptic long-term depression improves abnormal anxiety-like behavior in adult GluN2B-C456Y-mutant mice

Wangyong Shin[1☯], Kyungdeok Kim[1☯], Benjamin Serraz[2], Yi Sul Cho[3], Doyoun Kim[4], Muwon Kang[1], Eun-Jae Lee[5], Hyejin Lee[1], Yong Chul Bae[3], Pierre Paoletti[2], Eunjoon Kim[1,4]*

1 Department of Biological Sciences, Korea Advanced Institute of Science and Technology (KAIST), Daejeon, Korea, 2 Institut de Biologie de l'Ecole Normale Supérieure (IBENS), Ecole Normale Supérieure, Université PSL, CNRS, INSERM, Paris, France, 3 Department of Anatomy and Neurobiology, School of Dentistry, Kyungpook National University, Daegu, Korea, 4 Center for Synaptic Brain Dysfunctions, Institute for Basic Science (IBS), Daejeon, Korea, 5 Department of Neurology, Asan Medical Center, University of Ulsan, College of Medicine, Seoul, Korea

☯ These authors contributed equally to this work.
* kime@kaist.ac.kr

**Data Availability Statement:** All relevant data are within the paper and its Supporting Information files.

## Abstract

Extensive evidence links Glutamate receptor, ionotropic, NMDA2B (GRIN2B), encoding the GluN2B/NR2B subunit of N-methyl-D-aspartate receptors (NMDARs), with various neuro-developmental disorders, including autism spectrum disorders (ASDs), but the underlying mechanisms remain unclear. In addition, it remains unknown whether mutations in GluN2B, which starts to be expressed early in development, induces early pathophysiology that can be corrected by early treatments for long-lasting effects. We generated and characterized Grin2b-mutant mice that carry a heterozygous, ASD-risk C456Y mutation ($Grin2b^{+/C456Y}$). In $Grin2b^{+/C456Y}$ mice, GluN2B protein levels were strongly reduced in association with decreased hippocampal NMDAR currents and NMDAR-dependent long-term depression (LTD) but unaltered long-term potentiation, indicative of mutation-induced protein degradation and LTD sensitivity. Behaviorally, $Grin2b^{+/C456Y}$ mice showed normal social interaction but exhibited abnormal anxiolytic-like behavior. Importantly, early, but not late, treatment of young $Grin2b^{+/C456Y}$ mice with the NMDAR agonist D-cycloserine rescued NMDAR currents and LTD in juvenile mice and improved anxiolytic-like behavior in adult mice. Therefore, GluN2B-C456Y haploinsufficiency decreases GluN2B protein levels, NMDAR-dependent LTD, and anxiety-like behavior, and early activation of NMDAR function has long-lasting effects on adult mouse behavior.

## Introduction

Autism spectrum disorders (ASDs) are neurodevelopmental disorders characterized by social deficits and repetitive behaviors. Although a large number of ASD-risk mutations have been reported [1], the mechanisms underlying ASD remain largely unclear. An emerging ASD-

**Funding:** This work was supported by the European Research Council (ERC Advanced Grant #693021 to PP), the University Pierre-et-Marie-Curie (UPMC Paris 6) and the Fondation pour la Recherche Médicale (FRM) (fellowships to BS), the National Research Foundation of Korea (NRF) grant funded by the Korea government (MSIT, NRF-2017R1A5A2015391 to YCB), and the Institute for Basic Science (IBS-R002-D1 to EK). The funders had no role in study design, data collection and analysis, decision to publish, or preparation of the manuscript.

**Competing interests:** The authors have declared that no competing interests exist.

**Abbreviations:** AMPA, alpha-Amino-3-hydroxy-5-methyl-4-isoxazolepropionic acid; AMPAR, AMPA receptor; ASD, autism spectrum disorder; ATD, amino-terminal domain; CaMKII, $Ca^{2+}$/calmodulin dependent protein kinase II; DHPG, dihydroxyphenylglycine; E, embryonic day; EM, electron microscopic; EPSC, excitatory postsynaptic current; fEPSP, field excitatory postsynaptic potential; Grin1, *Glutamate receptor, ionotropic, NMDA1*; Grin2b, *Glutamate receptor, ionotropic, NMDA2B*; HEK-293, human embryonic kidney 293; HFS, high-frequency stimulation; i.p., intraperitoneal; IPSC, inhibitory postsynaptic current; KO, knockout; LBD, ligand-binding domain; LFS, low-frequency stimulation; LTD, long-term depression; LTP, long-term potentiation; mEPSC, miniature EPSC; mGluR, metabotropic glutamate receptor; mIPSC, miniature IPSC; mPFC, medial prefrontal cortex; NMDA, N-methyl-D-aspartate; NMDAR, NMDA receptor; ns, not significant; P, postnatal day; PSD, postsynaptic density; SC-CA1, Schaffer collateral-CA1 pyramidal; sEPSC, spontaneous EPSC; SFARI, Simons Foundation Autism Research Initiative; sIPSC, spontaneous IPSC; TBS, theta burst stimulation; USV, ultrasonic vocalization; WT, wild type.

related mechanism is dysfunction of N-methyl-D-aspartate (NMDA) receptors (NMDARs) [2], a key, multisubunit regulator of brain development and function that is subject to various forms of receptor modulation [3–7]. Many known ASD-risk genetic variants have been shown to cause NMDAR dysfunction in animal models of ASD [2] that are causally associated with ASD-like abnormal behaviors [8–10]. However, a better animal model of ASD for the NMDAR dysfunction hypothesis would presumably be one carrying mutations in *Glutamate receptor, ionotropic, NMDA1 (GRIN1); GRIN2A;* or *GRIN2B* genes encoding the main NMDAR subunits GluN1/NR1, GluN2A/NR2A, and GluN2B/NR2B, respectively.

Among known NMDAR subunit genes, *GRIN2B* is one of the most frequently mutated ASD-risk genes, belonging to category 1 in the Simons Foundation Autism Research Initiative (SFARI) gene list, and shows stronger impacts on ASD than mutations in *GRIN1* or *GRIN2A* [1,11–16]. In addition to ASD, *GRIN2B* has been extensively associated with various neurodevelopmental disorders, including developmental delay, intellectual disability, attention-deficit/hyperactivity disorder, epilepsy, schizophrenia, obsessive-compulsive disorder, and encephalopathy [5,15,17,18].

In line with the strong involvement of *GRIN2B* in diverse brain disorders, mice carrying a conventional homozygous deletion of *Grin2b* display impaired suckling, neonatal death during postnatal day (P) 1–3, and impaired hippocampal long-term depression (LTD) in neonates [19]. Similarly, a homozygous truncation of the intracellular C-terminal region of GluN2B causes perinatal lethality in mice [20]. These early studies were followed by those restricting homozygous *Grin2b* deletion to specific cell types and developmental stages to circumvent the strong developmental impacts of *Grin2b* deletion, which revealed the important roles of GluN2B in the regulation of long-term potentiation (LTP), LTD, and cognitive behaviors [21–23]. Notably, an early study investigated mice with heterozygous (not homozygous) *Grin2b* deletion and reported impaired LTP at the mutant hippocampal fimbrial-CA3 synapses [24], although associated behaviors were not investigated. Conversely, *Grin2b* overexpression has been shown to enhance LTP and learning and memory in mice [25]. These results suggest that GluN2B is important for normal brain development, synaptic plasticity, and cognitive behaviors.

However, dissimilar to the previous studies on *Grin2b* mice largely analyzing the synaptic and behavioral impacts of a homozygous *Grin2b* deletion, *GRIN2B* mutations identified in human brain disorders are preponderantly heterozygous mutations, and synaptic and behavioral phenotypes of heterozygous *Grin2b*-mutant mice remain largely unexplored. In addition, human *GRIN2B* mutations are often missense mutations that induce a single amino acid change in the encoded protein, again distinct from the null or truncation mutations previously studied in mice. Although many of the missense mutations of NMDAR subunits have been characterized in vitro [26], their in vivo impacts have been minimally studied.

In the present study, we generated and characterized a knock-in mouse line carrying an ASD-risk mutation (GluN2B-C456Y) in the *Grin2b* gene, a de novo mutation identified in a male individual with ASD and intellectual disability [12]. These heterozygous GluN2B-C456Y mutant mice (*Grin2b*+/C456Y) showed substantial decreases in GluN2B protein levels, suggestive of mutation-induced protein degradation in vivo. Currents of GluN2B-containing NMDARs and NMDAR-dependent LTD (but not LTP) were also decreased, revealing sensitivity of LTD to GluN2B haploinsufficiency. Behaviorally, these mice showed normal social interaction but enhanced anxiety-like behavior in pups and contrasting anxiolytic-like behavior in juveniles and adults. These synaptic and behavioral effects were largely mimicked by an independent mouse line carrying a conventional heterozygous *Grin2b* deletion (*Grin2b*+/−). Importantly, early, but not late, treatment of young mice (P7–16) with the NMDAR agonist D-cycloserine normalized NMDAR currents and LTD in juvenile *Grin2b*+/C456Y mice and

improved anxiolytic-like behavior in adult *Grin2b*<sup>+/C456Y</sup> mice, supporting the emerging concept in the field of neurodevelopmental and neuropsychiatric disorders that early and timely correction of key pathophysiological deficits is important for efficient and long-lasting beneficial effects.

## Results

### Structural and functional impacts of the GluN2B-C456Y mutation on GluN1/GluN2B receptors

Structural analysis has suggested that a missense mutation in the *GRIN2B* gene leading to a C456Y mutation in the GluN2B subunit of NMDARs disrupts a disulfide bond within a loop residing at the interface between the amino-terminal domain (ATD) and ligand-binding domain (LBD) [27]. In addition, experiments using *Xenopus* oocytes and human embryonic kidney 293 (HEK-293) cells have shown that the GluN2B-C456Y mutation induces multiple changes in the GluN2B protein, including protein degradation, limited surface trafficking, and gating alterations of GluN2B-containing NMDARs [27].

Our own structural investigation and functional characterization of the GluN2B-C456Y protein using *Xenopus* oocytes yielded overall similar results. Specifically, structural modeling, based on the known structure of GluN1/GluN2B NMDARs [28–30], revealed that the GluN2B-C456Y mutation in the LBD region alters the structure of a large loop protruding from the GluN2B LBD by disrupting the formation of an intraloop disulfide bond (S1A and S1B Fig). This loop, which is absent in alpha-Amino-3-hydroxy-5-methyl-4-isoxazolepropionic acid (AMPA) and kainate receptors, makes extensive intra- and intersubunit interactions within neighboring domains pointing to an important role in receptor assembly [28–30]. It is therefore likely that the GluN2B-C456Y mutation alters the receptor quaternary structure and in turn its function.

Diheteromers produced by coexpressing GluN2B-C456Y and wild-type (WT) GluN1 yielded NMDAR currents that were <1% of the currents produced by WT GluN1/GluN2B diheteromers (S2A Fig), thus revealing a strong expression phenotype, as previously observed [27]. Functional characterization of the small-amplitude mutant receptor currents showed an approximately 30% increase in receptor channel maximal open probability, as assessed by MK-801 inhibition kinetics [31], likely due to a decreased inhibition by ambient protons, as indicated by full proton dose-response curves (S2B and S2C Fig). In agreement with a decreased pH sensitivity, potentiation by spermine, a GluN2B-specific positive allosteric modulator [32], was also significantly reduced (S2D Fig). In addition, sensitivity to glycine was decreased, whereas sensitivities to glutamate and zinc, an endogenous allosteric inhibitor of NMDARs [33], were minimally affected (S2E–S2G Fig). Overall, these results indicate that the C456Y mutation in GluN2B drastically reduces expression and alter channel functions of recombinant NMDARs most likely because of GluN2B misfolding and degradation.

### The GluN2B-C456Y mutation decreases GluN2B and GluN1 protein levels and GluN2B-containing NMDAR currents in mice

To explore the impacts of the C456Y mutation on the stability or function of GluN2B in mice, we generated a knock-in mouse line carrying the C456Y mutation in the *Grin2b* gene (Fig 1A; S3A and S3B Fig). Homozygous C456Y-mutant (*Grin2b*<sup>C456Y/C456Y</sup>) mice showed neonatal death at P7 (approximately 0% survival), similar to the case for mice with a conventional homozygous *Grin2b* deletion [19,20]. In contrast, our heterozygous mutant (*Grin2b*<sup>+/C456Y</sup>) mice were produced with the expected mendelian ratios and showed normal survival and growth.

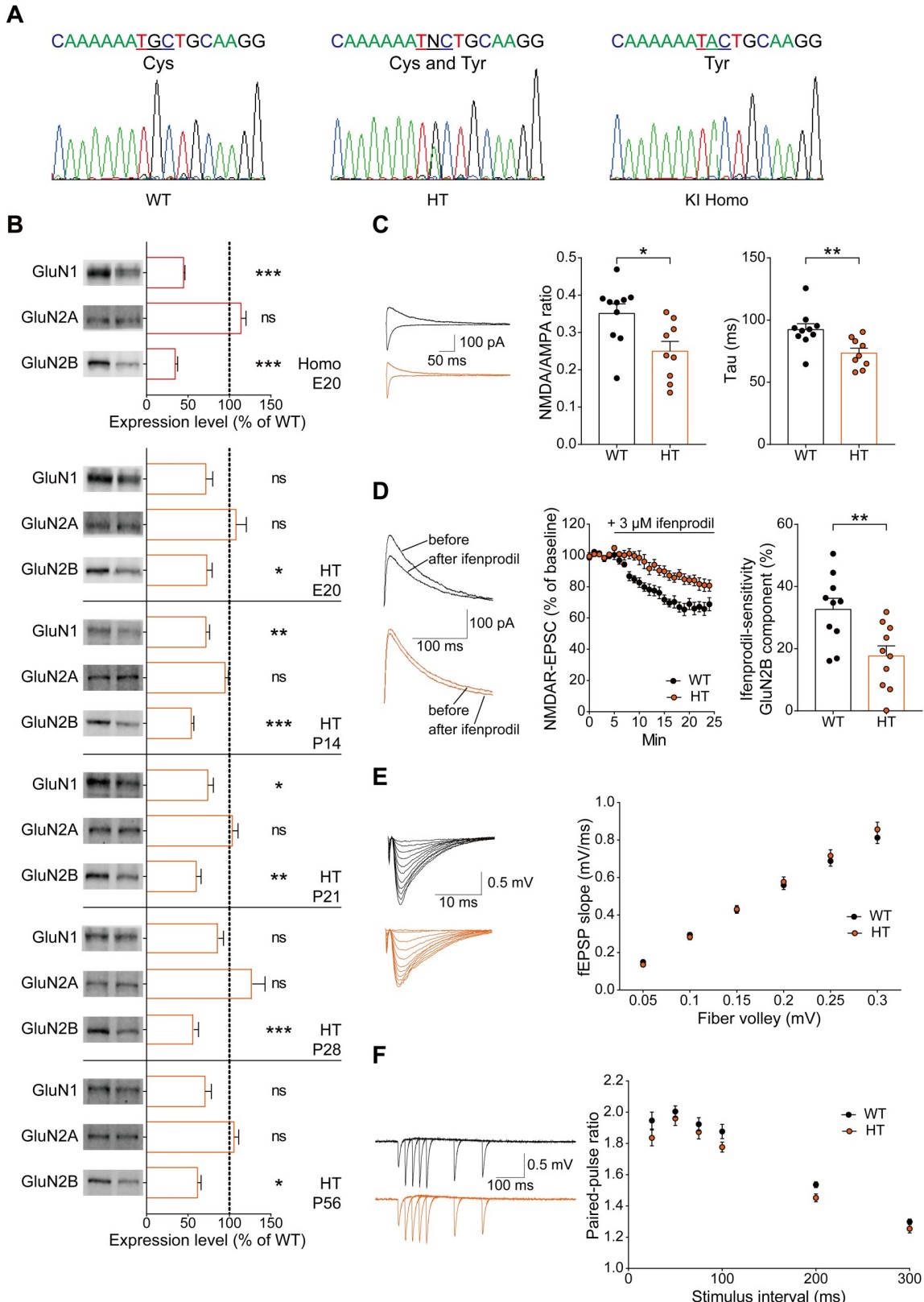

**Fig 1. The GluN2B-C456Y mutation leads to decreases in GluN2B and GluN1 protein levels and GluN2B-containing NMDAR currents in mice.** (A) Verification of the GluN2B-C456Y mutation in HT and homozygous ("Homo") KI mice, and validation of its

absence in WT mice, by DNA sequencing. (B) Decreased levels of GluN2B protein in the *Grin2b*+/C456Y brain. Whole-brain fractions from *Grin2b*+/C456Y mice at multiple developmental stages (E20, P14, P21, P28, and P56) were immunoblotted with anti-GluN1/2A/2B antibodies. Note that levels of GluN1 protein are also decreased, although to a lesser extent than those of GluN2B. For purposes of quantification, average levels of GluN1, Glu2A, and Glu2B proteins from *Grin2b*+/C456Y mice were normalized to those from WT mice. $n$ = 4 mice for WT and HT, $^*P < 0.05$, $^{**}P < 0.01$, $^{***}P < 0.001$, Student $t$ test. (C) Decreased ratio of evoked NMDAR- and AMPAR-mediated EPSCs (NMDA/AMPA ratio) at hippocampal SC-CA1 synapses of *Grin2b*+/C456Y mice (P19–23). Note the faster decay kinetics of the mutant NMDAR currents, indicative of a decrease in the GluN2B component. $n$ = 10 neurons from 5 mice for WT, and 9 (5) for HT, $^*P < 0.05$, $^{**}P < 0.01$, Student $t$ test. (D) Decreased proportion of ifenprodil-sensitive currents of GluN2B-containing NMDARs at SC-CA1 synapses of *Grin2b*+/C456Y mice (P21–23). $n$ = 10 neurons (7 mice) for WT and HT, $^{**}P < 0.01$, Student $t$ test. (E) Normal levels of basal excitatory synaptic transmission at SC-CA1 synapses of *Grin2b*+/C456Y mice (P27–41), as shown by the input-output relationship of evoked EPSCs. $n$ = 10 slices from 3 mice for WT and HT, one-way ANOVA. (F) Normal levels of paired-pulse facilitation at SC-CA1 synapses of *Grin2b*+/C456Y mice (P27–41). $n$ = 10 slices (3 mice) for WT and HT, one-way ANOVA. The numerical data underlying this figure can be found in S3 Data. AMPA, alpha-Amino-3-hydroxy-5-methyl-4-isoxazolepropionic acid; AMPAR, AMPA receptor; E, embryonic day; EPSC, excitatory postsynaptic current; fEPSP, field excitatory postsynaptic potential; HT, heterozygous; KI, knock-in; NMDA, N-methyl-D-aspartate; NMDAR, NMDA receptor; ns, not significant; P, postnatal day; SC-CA1, Schaffer collateral-CA1 pyramidal; WT, wild type.

Immunoblot analyses of whole-brain lysates and crude synaptosomal fractions from *Grin2b*+/C456Y mice at embryonic day 20 (E20) and several postnatal stages (P14, P21, P28, and P56) indicated approximately 30%–50% reductions in the levels of GluN2B protein (Fig 1B; S4A Fig). Notably, the levels of GluN1, but not GluN2A, were also reduced, although to a lesser extent than those of GluN2B, indicating that the stability of GluN1 strongly depends on GluN2B, whereas that of GluN2A does not. However, the C456Y mutation had no effect on mRNA levels of *Grin2b* or *Grin1* (encoding GluN1) (S4B Fig). These results indicate that the GluN2B-C456Y mutation induces a strong reduction in the levels of the GluN2B protein as well as a concomitant reduction in GluN1 protein levels in vivo, without affecting their mRNA levels.

We next tested whether the GluN2B-C456Y mutation affects NMDAR currents in the hippocampus of *Grin2b*+/C456Y mice. This mutation caused significant decreases in the ratio of NMDAR/AMPA receptor (AMPAR)-mediated evoked excitatory postsynaptic currents (EPSCs), tau of NMDAR current decay, and the amount of ifenprodil-sensitive current of GluN2B-containing NMDARs at Schaffer collateral-CA1 pyramidal (SC-CA1) synapses (Fig 1C and 1D). In contrast, *Grin2b*+/C456Y SC-CA1 synapses showed a normal input-output relationship of evoked EPSCs and paired-pulse facilitation (Fig 1E and 1F), indicative of normal AMPAR-mediated basal excitatory synaptic transmission and presynaptic release. These results suggest that the GluN2B-C456Y mutation selectively suppresses currents of GluN2B-containing NMDARs at hippocampal SC-CA1 synapses.

## The GluN2B-C456Y mutation reduces hippocampal NMDAR-dependent LTD without affecting LTP or mGluR-LTD

Previous studies on *Grin2b*-mutant mice have demonstrated the critical roles of GluN2B in the regulation of synaptic plasticity such as LTP and LTD [19–24], although the majority of these studies used mice carrying homozygous *Grin2b* deletion.

However, these observations are not congruent with human cases of *GRIN2B* mutations and related brain dysfunctions, in which heterozygous *GRIN2B* mutations are prevalent [1,11–17]. Thus, whether *Grin2b* haploinsufficiency in *Grin2b*-mutant mice would affect various forms of hippocampal synaptic plasticity or other synaptic functions is an important question that needs to be addressed. This question becomes more complicated when we consider the juvenile and adult stages, when both GluN2B and GluN2A are expressed and contribute to the formation of multiple forms of NMDARs with different subunit compositions, including diheteromeric (1/2A or 1/2B) and triheteromeric (1/2A/2B) NMDAR complexes [5,34,35].

To address this question, we measured several forms of synaptic plasticity in addition to low-frequency stimulation (LFS)-LTD, including LTP induced by high-frequency stimulation (HFS-LTP), theta burst stimulation–induced LTP (TBS-LTP), and metabotropic glutamate receptor (mGluR)-dependent LTD (mGluR-LTD) in the CA1 region of the *Grin2b*<sup>+/C456Y</sup> hippocampus at juvenile stages (P16–33).

The GluN2B-C456Y mutation reduced LFS-LTD by about 55% at *Grin2b*<sup>+/C456Y</sup> SC-CA1 synapses compared with WT mice (Fig 2A), a result similar to that obtained in neonatal mice with a conventional homozygous *Grin2b* deletion [19]. A similar decrease (approximately 84%) in LFS-LTD was observed in the prelimbic layer 1 region of the medial prefrontal cortex (mPFC) (Fig 2B). This result provides genetic evidence that LFS-LTD in the hippocampus is sensitive to *Grin2b* haploinsufficiency.

In contrast, the GluN2B-C456Y mutation had no effect on mGluR-LTD induced by the group I mGluR agonist dihydroxyphenylglycine (DHPG) at SC-CA1 synapses of *Grin2b*<sup>+/C456Y</sup> mice (Fig 2C). It also had no effect on HFS-LTP or TBS-LTP at *Grin2b*<sup>+/C456Y</sup> SC-CA1 synapses (Fig 2D and 2E). These results suggest that the heterozygous C456Y mutation and consequent decreases in GluN2B protein levels, GluN2B-dependent NMDAR currents, and LFS-LTD have no effect on other forms of synaptic plasticity in the hippocampus.

Because LTD is implicated in the regulation of synapse shrinkage and pruning [36], we attempted an electron microscopic (EM) analysis to see whether *Grin2b*<sup>+/C456Y</sup> mice display altered density or morphology of excitatory synapses. However, there were no genotype differences in the density and morphology (length, depth, and perforation [a measure of maturation]) of postsynaptic density (PSD) structures in the CA1 region of the WT and *Grin2b*<sup>+/C456Y</sup> hippocampus (P21) (Fig 2F), electron-dense multiprotein complexes at excitatory postsynaptic sites [37,38], suggesting that a moderate (approximately 50%) decrease in LTD does not induce morphological changes of excitatory synapses.

## The GluN2B-C456Y mutation does not affect spontaneous excitatory or inhibitory synaptic transmission or neuronal excitability

A previous study employed single-neuron gene deletion to show that GluN2A and GluN2B distinctly regulate the number and strength of functional excitatory synapses [39]. In addition, GluN2B is expressed in GABAergic interneurons [5] and NMDARs can function at presynaptic sites [40]. It is therefore possible that a heterozygous GluN2B-C456Y mutation might influence synaptic features unrelated to synaptic plasticity, such as synapse development and spontaneous synaptic transmission, at both excitatory and inhibitory synapses. In addition, mutations expected to mainly affect excitatory synapses are frequently associated with changes in intrinsic neuronal properties, such as neuronal excitability [41], suggesting that the GluN2B-C456Y mutation might also affect neuronal properties. To test these possibilities, we first measured spontaneous synaptic transmission at excitatory and inhibitory *Grin2b*<sup>+/C456Y</sup> synapses.

The frequency and amplitude of miniature EPSCs (mEPSCs) and miniature inhibitory postsynaptic currents (mIPSCs) did not differ in CA1 pyramidal neurons in the hippocampus of *Grin2b*<sup>+/C456Y</sup> mice compared with those of WT animals (S5A and S5B Fig), suggestive of normal development and efficacy of excitatory and inhibitory synapses. Moreover, there were no differences between genotypes in spontaneous EPSCs (sEPSCs) or spontaneous IPSCs (sIPSCs), measured in the absence of tetrodotoxin to allow network activities (S5C and S5D Fig), suggesting that excitatory network activity is unaltered in the hippocampus of *Grin2b*<sup>+/C456Y</sup> mice. We also measured the ratio of evoked EPSCs and IPSCs in the CA1 hippocampal region and found no genotype difference (S5E Fig).

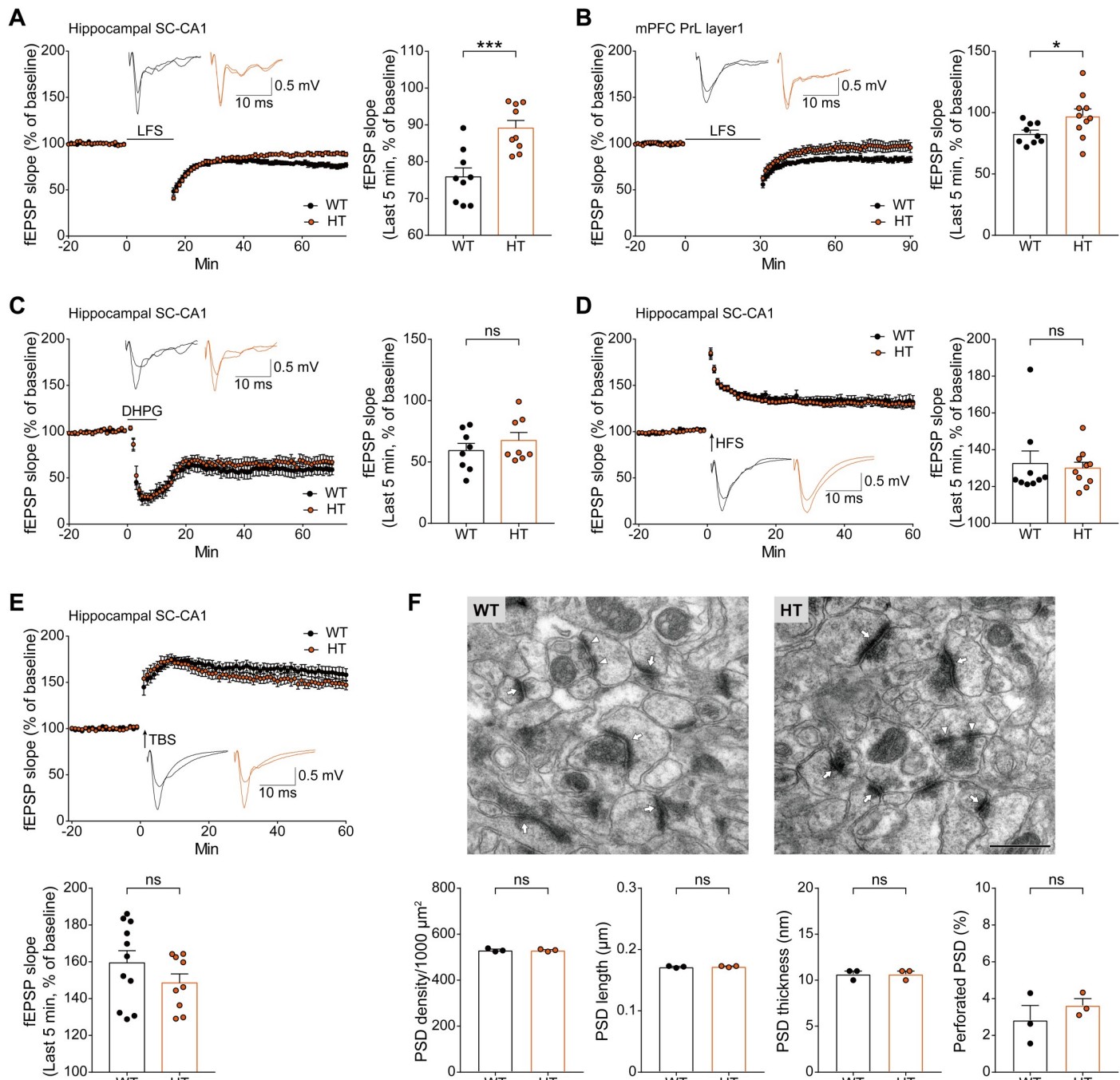

**Fig 2. Reduced LFS-LTD but normal mGluR-LTD, HFS-LTP, TBS-LTP, and PSD density and morphology in the *Grin2b+/C456Y* hippocampus.** (A) Reduced LFS-LTD at SC-CA1 synapses of *Grin2b+/C456Y* mice (P17–19). *n* = 9 neurons from 7 mice for WT (75.9% ± 2.4%), and 9 (5) for HT (89.1% ± 2.1%), ***P < 0.001, Student *t* test. (B) Reduced LFS-LTD in the PrL 1 region of the mPFC in *Grin2b+/C456Y* mice (P17–20). *n* = 9 neurons from 5 mice for WT (82.9% ± 2.8%), and 10 (5) for HT (97.3% ± 5.8%), *P < 0.05, Student *t* test. (C) Normal mGluR-LTD induced by the group I mGluR agonist DHPG (50 μM for 10 minutes) at SC-CA1 synapses of *Grin2b+/C456Y* mice (P16–20). *n* = 8 (6) for WT (59.3% ± 5.8%), and 8 (4) for HT (67.6% ± 6.4%), Student *t* test. (D) Normal HFS-LTP at SC-CA1 synapses of *Grin2b+/C456Y* mice (P27–33). *n* = 9 (5) for WT (132.5% ± 6.8%), and 10 (4) for HT (130.0% ± 3.2%), Mann–Whitney test. (E) Normal TBS-LTP at SC-CA1 synapses of *Grin2b+/C456Y* mice (P28–32). *n* = 11 (6) for WT (159.4% ± 6.6%), and 9 (4) for HT (148.5% ± 4.9%), Student *t* test. (F) Normal PSD density and morphology (length, depth, and perforation) in the CA1 region of *Grin2b+/C456Y* mice (P21). *n* = 3 mice for WT and HT, Student *t* test. The numerical data underlying this figure can be found in S3 Data. DHPG, dihydroxyphenylglycine; fEPSP, field excitatory postsynaptic potential; HFS, high-frequency stimulation; HT, heterozygous; LTD, long-term depression; LFS, low-frequency stimulation; LTP, long-term potentiation; mGluR, metabotropic glutamate receptor; mPFC, medial prefrontal cortex; ns, not significant; P, postnatal day; PrL, prelimbic layer; PSD, postsynaptic density; SC-CA1, Schaffer collateral-CA1 pyramidal; TBS, theta burst stimulation; WT, wild type.

In addition to spontaneous and evoked synaptic transmission, neural excitability was unaltered in *Grin2b*<sup>+/C456Y</sup> CA1 pyramidal neurons, as shown by current-firing curves (S5F Fig). These results collectively suggest that, in contrast to its effects on LFS-LTD, the heterozygous GluN2B-C456Y mutation does not affect neuronal excitability or excitatory or inhibitory synapse development or function in the hippocampus in the presence or absence of network activity.

## *Grin2b*<sup>+/C456Y</sup> mice display hypoactivity, anxiolytic-like behavior, and moderate repetitive self-grooming

To explore behavioral impacts of the GluN2B-C456Y mutation, we subjected *Grin2b*<sup>+/C456Y</sup> mice to a battery of behavioral tests. Adult male *Grin2b*<sup>+/C456Y</sup> mice displayed hypoactivity in the open-field test (Fig 3A and 3B) but spent normal amounts of time in the center region of the open-field arena, indicative of largely normal anxiety-like behavior (Fig 3C and 3D). These mice, however, displayed anxiolytic-like behavior during the first 10 minutes in the arena, likely reflecting a modified response to a novel environment.

In the elevated plus-maze, *Grin2b*<sup>+/C456Y</sup> mice also showed anxiolytic-like behavior, as shown by both the number of entries into and time spent in closed/open arms (Fig 3E–3H). In contrast, these mice showed normal anxiety-like behaviors in the light-dark test, as shown by time spent in the light chamber (Fig 3I). These results suggest that *Grin2b*<sup>+/C456Y</sup> mice display anxiolytic-like behavior in the elevated plus-maze.

In tests measuring learning and memory, *Grin2b*<sup>+/C456Y</sup> mice showed normal levels of learning and memory in the learning and probe phases of both initial- and reversal-learning sessions of the Morris water maze (S6A–S6E Fig). In addition, they showed a normal preference for a novel object over a familiar object in the novel object–recognition test (S6F Fig).

Contrary to our expectation, *Grin2b*<sup>+/C456Y</sup> mice showed largely normal social behaviors, including social approach and social novelty recognition in the three-chamber test [42]; social interaction between freely moving mice in the direct social-interaction test; and ultrasonic vocalizations (USVs), a form of social communication in rodents, upon encountering a female (S7A–S7N Fig) [42–44].

Furthermore, these mice showed enhanced self-grooming (but normal digging) in home cages with bedding but showed no repetitive self-grooming in a novel chamber without bedding (S7O–S7Q Fig), indicative of a moderate increase in self-grooming. These results collectively suggest that the GluN2B-C456Y mutation leads to hypoactivity, anxiolytic-like behavior, and moderately enhanced self-grooming, without affecting social interaction, social communication, or learning and memory in mice.

## A conventional heterozygous *Grin2b* deletion in mice leads to hypoactivity and anxiolytic-like behavior

We next employed an independent mouse line carrying a conventional heterozygous *Grin2b* deletion (*Grin2b*<sup>+/−</sup>) to see whether the behavioral phenotypes observed in *Grin2b*<sup>+/C456Y</sup> mice could be reproduced. This mouse line has been used previously to demonstrate that a homozygous null *Grin2b* mutation entirely eliminates GluN2B protein and causes severe phenotypes, including impaired suckling and neonatal death [19].

*Grin2b*<sup>+/−</sup> mice showed decreased (approximately 50%) whole-brain levels of the GluN2B, but not GluN1 or GluN2A, subunit of NMDARs at P14 and P21 (S8A and S8B Fig), partly similar to the results from *Grin2b*<sup>+/C456Y</sup> mice in which both GluN2B and GluN1 levels were decreased (P14 and P21). Behaviorally, adult male *Grin2b*<sup>+/−</sup> mice showed phenotypes that were largely similar to those observed in adult male *Grin2b*<sup>+/C456Y</sup> mice, including hypoactivity

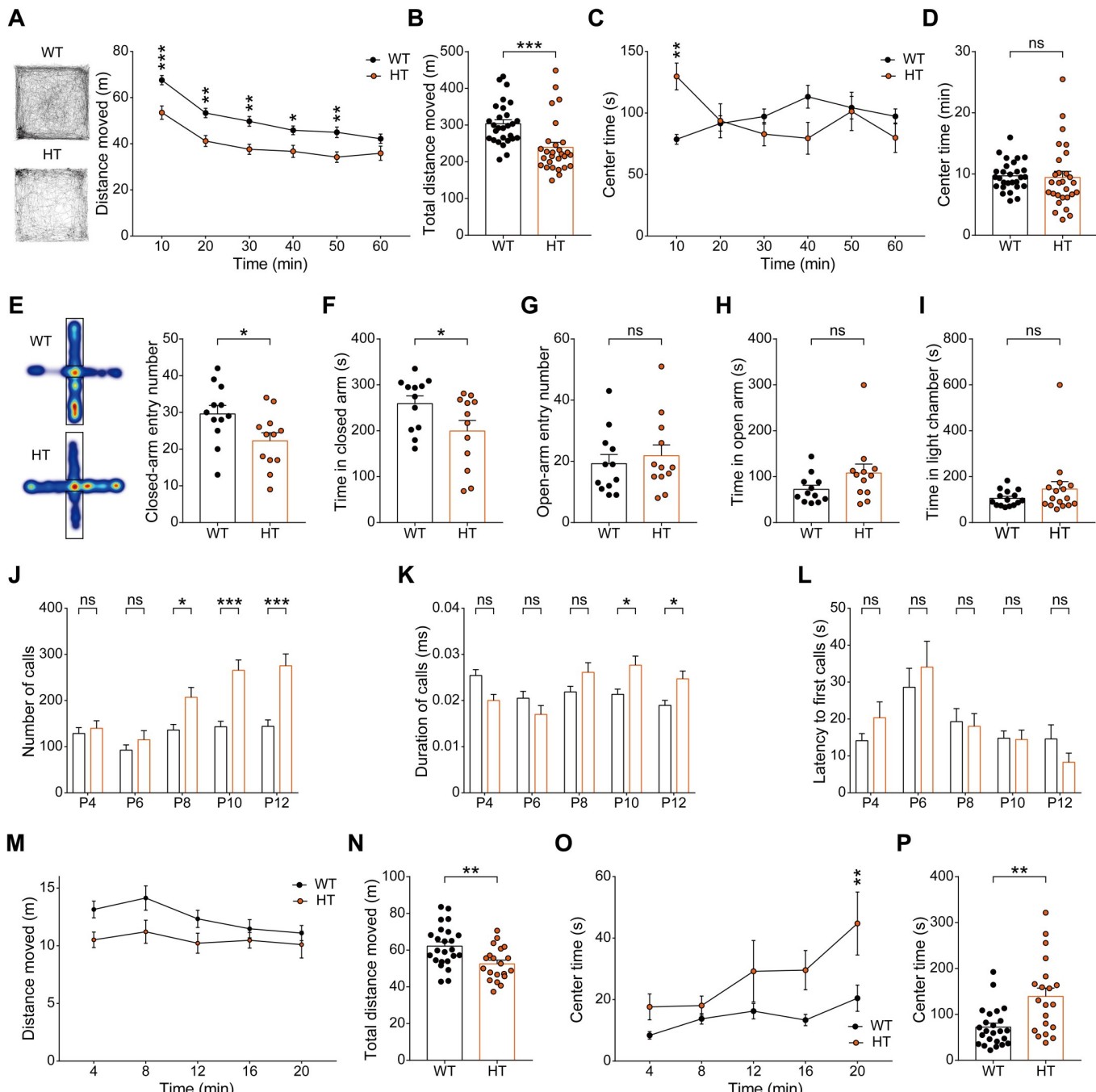

**Fig 3. Adult and juvenile *Grin2b+/C456Y* mice display hypoactivity and anxiolytic-like behavior, whereas *Grin2b+/C456Y* pups show anxiety-like enhanced USVs upon mother separation.** (A–D) Hypoactivity, but normal anxiety-like behavior, in *Grin2b+/C456Y* mice (P68–78) in the open-field test, as shown by distance moved and time spent in the center region of the open-field arena. $n = 28$ mice for WT and HT, $^*P < 0.05$, $^{**}P < 0.01$, $^{***}P < 0.001$, two-way ANOVA with Sidak's test and Mann-Whitney test. (E–H) Anxiolytic-like behavior in *Grin2b+/C456Y* mice (P70–124) in the elevated plus-maze, as shown by entries into and time spent in closed/open arms. $n = 12$ mice for WT and HT, $^*P < 0.05$, Student $t$ test (time in closed arms and closed-arm entry number) and Mann-Whitney test (time in open arms and open-arm entry number). (I) Normal anxiety-like behavior in *Grin2b+/C456Y* mice (P75) in the light-dark test, as shown by time spent in the light chamber. $n = 15$ mice for WT and 16 for HT, Mann-Whitney test. (J–L) *Grin2b+/C456Y* pups (P4–12) emit strongly increased USVs upon mother separation, as shown by the total number of USV calls, duration of each USV calls, and latency to first calls. $n = 62$ (WT) and 38 (HT), $^*P < 0.05$, $^{***}P < 0.001$, two-way ANOVA with Sidak's test. (M–P) *Grin2b+/C456Y* juveniles (P19–21) show modest hypoactivity and anxiolytic-like behavior in the open-field test, as shown by distance moved and time spent in the center region of the open-field arena. $n = 24$ mice for WT and 20 for HT, $^{**}P < 0.01$, two-way ANOVA with Sidak's test, Student $t$ test (distance moved) and Mann-Whitney test (center time). The numerical data underlying this figure can be found in S3 Data. HT, heterozygous; ns, not significant; P, postnatal day; USV, ultrasonic vocalization; WT, wild type.

and moderately anxiolytic-like behavior (S8C–S8K Fig), as well as normal social interaction and communication and object-recognition memory (S9A–S9K and S9N Fig). Unlike *Grin2b*+/C456Y mice, which showed modestly enhanced self-grooming, *Grin2b*+/− mice showed no repetitive self-grooming (S9L and S9M Fig).

These results indicate that the heterozygous C456Y mutation and conventional *Grin2b* heterozygosis lead to similar, although not identical, biochemical and behavioral phenotypes and suggest that the phenotypes observed in *Grin2b*+/C456Y mice are likely consequences of the loss (not gain) of GluN2B function. The small differences in the behaviors of the two mouse lines may reflect minor effects attributable to the specific mutation/deletion in the *Grin2b* gene.

## *Grin2b*+/C456Y pups show anxiety-like behavior whereas *Grin2b*+/C456Y juveniles show normal or anxiolytic-like behavior

ASD is characterized by early onset of core and comorbid symptoms. When *Grin2b*+/C456Y pups (P4–12) were tested for the emission of USVs upon mother separation, a measure of anxiety in rodents responsive to anxiolytic medications [45], these pups showed strongly enhanced USVs, as determined by the total number of USV calls, duration of each USV calls, and latency to first calls (Fig 3J–3L). This suggests that *Grin2b*+/C456Y pups display anxiety-like behaviors, similar to the anxiety symptoms comorbid with human ASD [46].

When *Grin2b*+/C456Y juveniles (P18–26) were subjected to a battery of behavioral tests, they displayed hypoactivity, similar to adult mice, and, notably, strong anxiolytic-like behavior in the center region of the open-field arena (Fig 3M–3P). However, upon mother separation and reunification in a mother-homing test, *Grin2b*+/C456Y juveniles spent normal amounts of time with the reunited mothers (S10A and S10B Fig), suggestive of normal anxiety-like behaviors. Therefore, anxiety-like behavior in *Grin2b*+/C456Y pups seems to be strongly weakened or changed into anxiolytic-like behavior at juvenile stages, similar to the anxiolytic-like behaviors in adults.

*Grin2b*+/C456Y juveniles showed normal social interaction, as shown by the juvenile play test (S10C Fig), similar to adult mice. In addition, these mice showed normal self-grooming and digging in home cages with bedding (S10D and S10E Fig), partly dissimilar to the adult mutant mice that show enhanced self-grooming but normal digging in home cages with bedding. These results suggest that self-grooming in *Grin2b*+/C456Y mice develops slowly in late life after the juvenile stage.

## Early correction of NMDAR function and NMDAR-dependent LTD by D-cycloserine improves anxiolytic-like behavior in adult *Grin2b*+/C456Y mice

The reduced NMDAR function and LTD observed in young (2–3-week-old) *Grin2b*+/C456Y mice might be associated with the behavioral abnormalities (hypoactivity and anxiolytic-like behavior) observed in adult (2–4-month-old) *Grin2b*+/C456Y mice. This hypothesis could be tested by normalizing the reduced NMDAR function and NMDAR-dependent LTD in early stages and examining whether these corrections are associated with behavioral rescues at late stages. To this end, we used D-cycloserine, a partial agonist at the glycine-binding site of NMDARs with increasing potential for the treatment of neurological and neuropsychiatric disorders [47].

We first tested whether the reduced LFS-LTD is attributable to decreased NMDAR currents in *Grin2b*+/C456Y mice. In hippocampal slices from young *Grin2b*+/C456Y mice, application of D-cycloserine (10 μM), which can still activate mutant GluN2B-C456Y receptors (S2H Fig), fully normalized the reduced LFS-LTD at *Grin2b*+/C456Y SC-CA1 synapses, without affecting WT synapses (Fig 4A). These results suggest that abnormal NMDAR currents are associated with reduced LFS-LTD at *Grin2b*+/C456Y hippocampal SC-CA1 synapses in juvenile mice.

We next attempted early, chronic treatment of young *Grin2b*$^{+/C456Y}$ mice with D-cycloserine (40 mg/kg), administered orally twice daily for 10 days (P7–16), followed by measurements of NMDA/AMPA ratio, paired-pulse facilitation, and LFS-LTD in juvenile mice (P17–21) and behavioral experiments in adult mice (>P56) (Fig 4B). Early D-cycloserine treatment fully normalized the reduced NMDA/AMPA ratio and LFS-LTD at *Grin2b*$^{+/C456Y}$ SC-CA1 synapses in juvenile mice (P16–21), without affecting paired-pulse facilitation (Fig 4C–4E). WT synapses were unaffected in these measurements.

In addition, early D-cycloserine treatment improved anxiolytic-like behavior in *Grin2b*$^{+/C456Y}$ mice in the elevated plus-maze test, without affecting WT mice (Fig 4F–4I). In contrast, D-cycloserine had no effect on hypoactivity in *Grin2b*$^{+/C456Y}$ mice (Fig 4J–4M). We could not test whether early D-cycloserine treatment (P7–16) could affect pup USVs in *Grin2b*$^{+/C456Y}$ mice because the time window for the treatment fell after the early time window for pup USV (P4–12).

When *Grin2b*$^{+/-}$ mice carrying conventional heterozygous *Grin2b* deletion were tested for D-cycloserine-dependent rescue of synaptic and behavioral deficits, early chronic D-cycloserine treatment (P7–16, 40 mg/kg, twice daily for 10 days; oral) normalized the decreased LFS-LTD at *Grin2b*$^{+/C456Y}$ SC-CA1 synapses, without affecting WT synapses (S11A and S11B Fig). Early D-cycloserine treatment also had no effect on the hypoactivity of *Grin2b*$^{+/-}$ mice (S11C Fig), similar to the results from *Grin2b*$^{+/C456Y}$ mice. We could not determine whether D-cycloserine has any effect on the anxiety-like behavior in *Grin2b*$^{+/-}$ mice (S11C Fig) because the anxiolytic-like behavior in *Grin2b*$^{+/-}$ mice was weaker than that in *Grin2b*$^{+/C456Y}$ mice, and the small baseline difference between WT and *Grin2b*$^{+/-}$ mice became insignificant (between vehicle-treated WT and *Grin2b*$^{+/-}$ mice) by the chronic drug treatment procedures (oral, twice/day for 10 days).

## Late NMDAR activation by D-cycloserine does not improve anxiolytic-like behavior or hypoactivity in adult *Grin2b*$^{+/C456Y}$ mice

*Grin2b*$^{+/C456Y}$ mice display decreased GluN2B levels at an adult stage (P56), suggesting that the continuing decrease in GluN2B levels in adult mutant mice, in addition to the reduced LFS-LTD in young mutant mice, might be associated with anxiety-like behavior or hypoactivity. To test this, we attempted to enhance NMDAR function in adult *Grin2b*$^{+/C456Y}$ mice by treating with D-cycloserine. In these experiments, we first used an acute treatment paradigm because acute D-cycloserine treatment has been previously shown to rescue ASD-like behaviors in many mouse models of ASD [9,10,48–50].

In contrast to early chronic treatment, late acute D-cycloserine treatment (20 mg/kg; intraperitoneal [i.p.]) in the adult stage had no effect on anxiolytic-like behavior or hypoactivity in *Grin2b*$^{+/C456Y}$ mice in open-field or elevated plus-maze tests (Fig 5A–5I); it also had no effect on WT mice. In addition, late chronic D-cycloserine treatment (P57–66, 40 mg/kg, twice/day, oral) had no effect on the anxiolytic-like behavior or hypoactivity in *Grin2b*$^{+/C456Y}$ mice (S12 Fig). However, the late chronic drug treatment procedures using a restrainer substantially blunted the baseline differences in elevated plus-maze variables (but not locomotion) between vehicle-treated WT and mutant mice, making it difficult to assess the drug effects on these values. These results collectively suggest that late treatment of *Grin2b*$^{+/C456Y}$ mice with D-cycloserine to enhance NMDAR function has no effect on anxiolytic-like behavior or hypoactivity, highlighting the importance of early treatments.

## Discussion

In this study, we demonstrated that mice carrying a heterozygous C456Y mutation in the GluN2B subunit of NMDARs, an ASD-risk mutation in humans, exhibit decreased GluN2B and GluN1

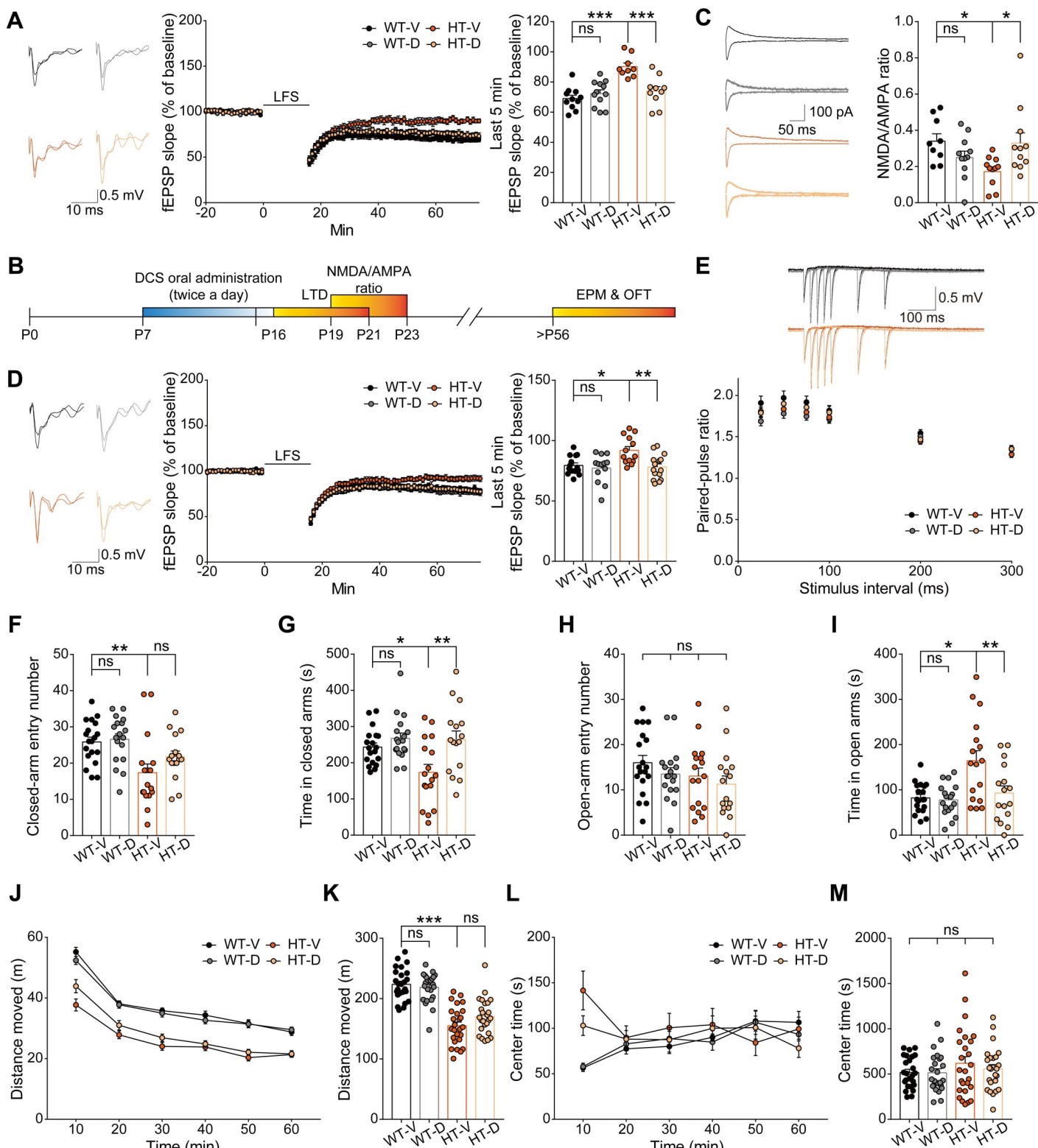

**Fig 4. Early correction of NMDAR function and NMDAR-dependent LTD by DCS treatment improves anxiolytic-like behavior, but not hypoactivity, in adult Grin2b+/C456Y mice.** (A) Acute treatment with 10 μM DCS normalizes LFS-LTD at SC-CA1 synapses in hippocampal slices from juvenile Grin2b+/C456Y mice (P17–19) without affecting WT synapses. n = 11 cells (4 mice) for WT_V (69.0% ± 7.7%), 12 (4) for WT_D (72.4% ± 2.4%), 9 (4) for HT_V (90.2% ± 2.4%), 10 (4) for HT_D (73.9% ± 3.1%), ***P < 0.001, two-way ANOVA with Tukey's test. (B) Experimental strategy for chronic, oral DCS treatment (40 mg/kg), twice daily for 10 days (P7–

16), in young Grin2b$^{+/C456Y}$ mice followed by measurements of NMDA/AMPA ratio, paired-pulse facilitation, and LTD in juvenile mice (P17–21) and behavioral tests (EPM and OFT) in adult mice (>P56). (C) Early chronic oral DCS treatment (40 mg/kg) normalizes the decreased NMDA/AMPA ratio at SC-CA1 synapses in juvenile Grin2b$^{+/C456Y}$ mice (P19–23). n = 9 cells from 4 mice for WT_V, 11 (5) for WT_D, 11 (5) for HT_V, 11 (3) for HT_D, *P < 0.05, two-way ANOVA with Tukey's test. (D) Early chronic oral DCS treatment (40 mg/kg) normalizes LFS-LTD at SC-CA1 synapses in juvenile Grin2b$^{+/C456Y}$ mice (P17–20). n = 14 slices from 5 mice for WT_V (79.5% ± 2.0%), 13 (6) for WT_D (77.2% ± 3.2%), 13 (8) for HT_V (92.0% ± 3.2%), 15 (7) for HT_D (78.1% ± 2.6%), *P < 0.05, **P < 0.01, two-way ANOVA with Tukey's test. (E) Early chronic oral DCS treatment (40 mg/kg) does not affect paired-pulse facilitation at SC-CA1 synapses in juvenile Grin2b$^{+/C456Y}$ mice (P26–28). n = 12 (3) for WT_V, 12 (3) for WT_D, 11 (3) for HT_V, 12 (3) for HT_D, two-way ANOVA with Tukey's test. (F–I) Early chronic oral DCS treatment (40 mg/kg) improves anxiolytic-like behavior in adult Grin2b$^{+/C456Y}$ mice (P63–73). n = 19 mice for WT_D, 18 for WT_D, 17 for HT_V, 16 for HT_D, *P < 0.05, **P < 0.01, two-way ANOVA with Tukey's test. (J–M) Early chronic oral DCS treatment (40 mg/kg) has no effect on hypoactivity in adult Grin2b$^{+/C456Y}$ mice (P60–71). Note that early DCS treatment did not affect the time spent in the center by WT or mutant mice. n = 25 mice for WT_D, 24 for WT_D, 26 for HT_V, 26 for HT_D, ***P < 0.001, two-way ANOVA with Tukey's test. The numerical data underlying this figure can be found in S3 Data. AMPA, alpha-Amino-3-hydroxy-5-methyl-4-isoxazolepropionic acid; DCS, D-cycloserine; EPM, elevated plus-maze; fEPSP, field excitatory postsynaptic potential; HT, heterozygous; HT_D, HT with DCS; HT_V, HT with vehicle; LFS, low-frequency stimulation; LTD, long-term depression; NMDA, N-methyl-D-aspartate; NMDAR, NMDA receptor; ns, not significant; OFT, open-field test; P, postnatal day; SC-CA1, Schaffer collateral-CA1 pyramidal; WT, wild type; WT_D, WT with DCS; WT_V, WT with vehicle.

protein levels, diminished currents of GluN2B-containing NMDARs, and reduced LFS-LTD. This mutation also induced anxiolytic-like behavior that can be corrected by early, but not late, D-cycloserine treatment that restores NMDAR function and NMDAR-dependent LTD.

## C456Y mutation and GluN2B proteins

A key finding in our study is that the GluN2B-C456Y mutation induces substantial degradation of the GluN2B protein in mice. This conclusion is supported by the measurement of

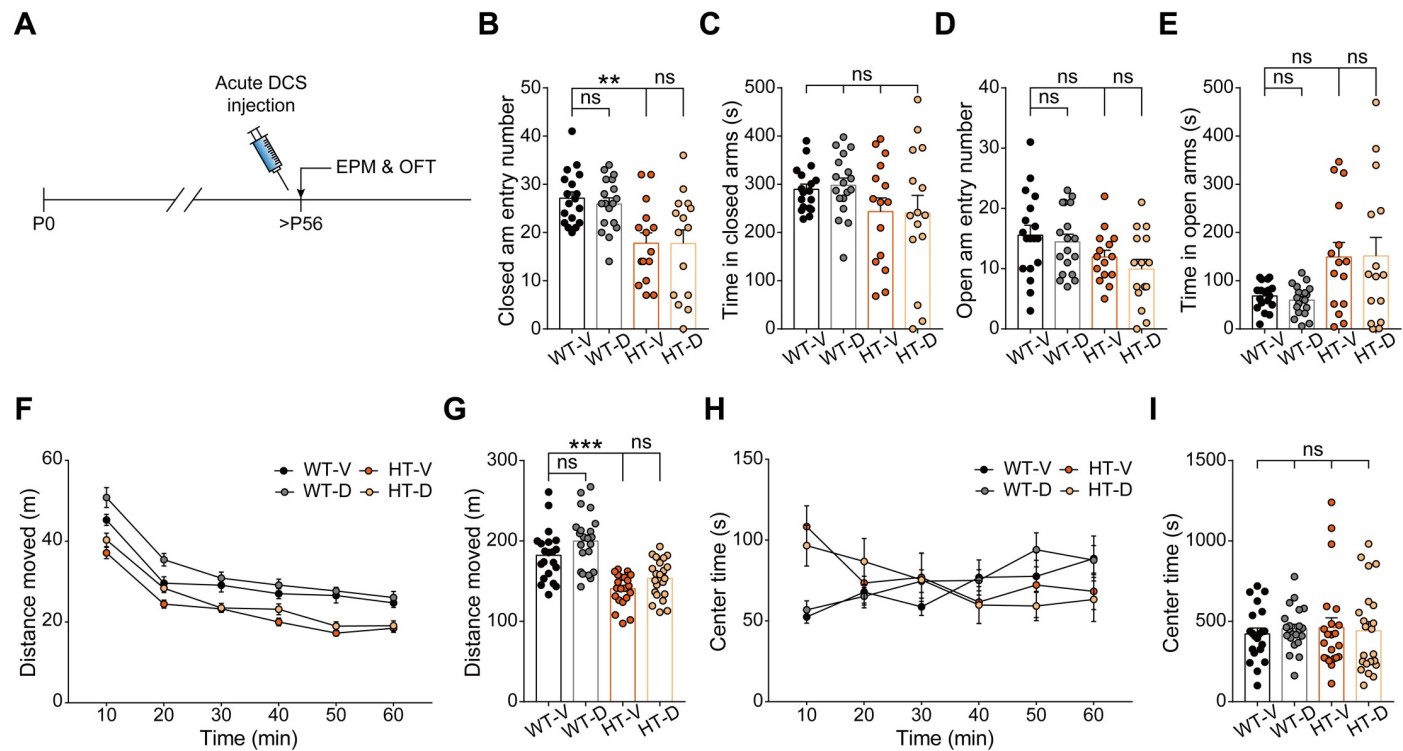

**Fig 5. Late acute DCS treatment has no effect on anxiolytic-like behavior or hypoactivity in adult Grin2b$^{+/C456Y}$ mice.** (A–E) Acute DCS treatment (20 mg/kg; single i.p. injection 30 minutes prior to the experiment) in adult mice (P62–76) does not affect anxiolytic-like behavior in the EPM. n = 18 mice for WT_D, 18 for WT_D, 15 for HT_V, 15 for HT_D, **P < 0.01, two-way ANOVA with Tukey's test. (F–I) Acute DCS treatment (20 mg/kg; single i.p. injection 30 minutes prior to the experiment) in adult Grin2b$^{+/C456Y}$ mice (P59–73) does not affect hypoactivity in the OFT. n = 21 mice for WT_D, 22 for WT_D, 22 for HT_V, 22 for HT_D, ***P < 0.001, two-way ANOVA with Tukey's test. The numerical data underlying this figure can be found in S3 Data. DCS, D-cycloserine; EPM, elevated plus-maze; HT, heterozygous; HT_D, HT with DCS; HT_V, HT with vehicle; i.p., intraperitoneal; ns, not significant; OFT, open-field test; P, postnatal day; WT, wild type; WT_D, WT with DCS; WT_V, WT with vehicle.

GluN2B protein levels in the *Grin2b+/C456Y* brain at various developmental stages. A previous study on multiple GluN2A/B mutations using structural analysis and oocyte/HEK cell experiments reported similar findings on the impact of the GluN2B-C456Y mutation [27]. Interestingly, a similar expression phenotype was observed for the patient-derived GluN2A-C436R mutation, which also disrupts a disulfide bond within LBD loop 1 [31]. Our study extends these previous findings by providing in vivo evidence of the importance of the C456Y mutation and the proper folding of LBD loop 1 for GluN2B protein levels.

Our results further reveal an impact of reduced GluN2B protein levels on GluN1 protein levels, although the magnitude of this latter decrease was less than that of GluN2B. This further supports the previously reported importance of GluN2B in the maintenance of normal levels of GluN1 [21,23]. This decrease in the GluN1 subunit in our study does not seem to involve changes in *Grin1* mRNA levels. It may occur because the reduction in GluN2B protein levels may lead to a situation in which some GluN1 proteins that can no longer associate with GluN2B to form heteromeric NMDAR complexes become destabilized and degraded. It is possible that some GluN1 proteins may fail to associate with mutant GluN2B proteins beginning in the endoplasmic reticulum [7] and are degraded via the ubiquitin-proteasomal pathway following retrograde transport to the cytoplasm [51–54]. Alternatively, the two proteins may initially associate with each other and reach the plasma membrane surface and synaptic sites but gradually dissociate from one another, leaving GluN1 subject to endocytosis and degradation through the late endosomal-lysosomal pathway involving a conserved membrane-proximal signal present in GluN1 [55,56].

Complicating the situation is the fact that GluN2B can form a triheteromeric complex with GluN1 and GluN2A [5,34,35,57,58] that is known to be the major NMDAR population in the adult hippocampus [39,59,60]. Although further details remain to be elucidated, the concomitant reduction in GluN1 levels creates a situation in which GluN1 protein is produced normally but not used.

In addition, given that diheteromic and triheteromeric NMDAR complexes display distinct biophysical and pharmacological properties in different spatiotemporal contexts [35,61–63], the reduced levels of GluN2B and GluN1 in *Grin2b+/C456Y* mice would affect both diheteromic and triheteromeric NMDAR complexes differentially in different brain regions, cell types, and developmental stages.

## GluN2B-C456Y and LTD

Another key finding of our study is the reduced NMDAR-dependent LFS-LTD by about 50% at *Grin2b+/C456Y* hippocampal SC-CA1 synapses. Previous studies on genetic *Grin2b* deletion and its impacts on LTD found near-complete impairments of LTD in neonate mice (P1–3) carrying a conventional homozygous *Grin2b* deletion [19] and in adult mice (14–22 weeks) carrying conditional homozygous *Grin2b* deletion restricted to Ca$^{2+}$/calmodulin dependent protein kinase II (CaMKII)-positive principal neurons in the cortex and hippocampal CA1 region [21]. The design of our *Grin2b*-mutant mouse study differs from those of the previous studies in the following respects: (1) use of a patient-derived knock-in mutation rather than a conventional, or conditional, gene deletion; (2) use of heterozygous instead of homozygous mutant mice (an early study on heterozygous *Grin2b* mice examined only LTP but not LTD [24]); and (3) analysis of LTD at a juvenile stage rather than at a neonatal or adult stage. In addition, our results indicate that the heterozygous GluN2B-C456Y mutation has no effect on other synaptic and neuronal variables, such as spontaneous synaptic transmission in CA1 neurons (mEPSCs, mIPSCs, sEPSCs, sIPSCs), basal transmission at SC-CA1 synapses (evoked EPSCs), the ratio of evoked IPSCs and EPSCs, and intrinsic neuronal excitability of CA1

neurons. Moreover, Grin2b$^{+/C456Y}$ hippocampal SC-CA1 synapses displayed normal HFS-LTP, TBS-LTP, or mGluR-LTD. The lack of changes in LTP (HFS and TBS) measured during a late juvenile stage (P27–33) could be because the postnatal switch from GluN2B to GluN2A by neuronal activity may be largely complete at this stage [5]. Together, these results support the established notion that genetic deletion of Grin2b suppresses LTD and extends it by demonstrating that the patient-derived heterozygous GluN2B-C456Y mutation induces a selective reduction in LTD by approximately 50% in juvenile mice.

A straightforward mechanism underlying the decreased LFS-LTD at Grin2b$^{+/C456Y}$ hippocampal SC-CA1 synapses would be decreased GluN2B function. Previous studies employing pharmacological inhibitors of GluN2B, however, yielded conflicting results, with their significant effects on LTD [64–67] or insignificant effects on LTD [68,69] (reviewed in [70]). This difference could be attributable to multiple factors, including the limited selectivity of the GluN2B inhibitors [71], differential actions of GluN2B inhibitors on di- and triheteromeric NMDARs [61,62], and influences of GluN2B inhibitors on glutamate dissociation rate [39,72,73].

Notably, a recent study employing single-neuron gene knockout (KO) has reported that GluN2A or GluN2B is not critically required for ionotropic or non-ionotropic (not involving NMDAR-dependent ion flow [74–76]) NMDAR-dependent LTD, whereas GluN1 is required for non-ionotropic NMDAR-dependent LTD [77]. It is thus possible that the reduced levels of GluN1 in Grin2b$^{+/C456Y}$ mice may contribute to the reduced LTD at SC-CA1 synapses. However, the previous single-neuron KO study employed AAV-dependent gene KO at the mouse age of P0–1, leaving GluN2B expression and function at embryonic stages unaffected. In addition, the single-neuron KO study would lead to homozygous (not heterozygous) Grin2b deletion, which might also affect the results.

## GluN2B-C456Y and behaviors

Grin2b$^{+/C456Y}$ mice showed moderately enhanced self-grooming, a core ASD-like behavior, in home cages with bedding, but normal self-grooming in a novel chamber without bedding, suggesting that these mice display moderately enhanced self-grooming that is suppressed by a novel environment. Moreover, enhanced self-grooming in home cages with bedding was observed in adult but not juvenile Grin2b$^{+/C456Y}$ mice, suggesting that repetitive self-grooming develops late in life in Grin2b$^{+/C456Y}$ mice and thus is unlikely to be ameliorated by early D-cycloserine treatment.

Contrary to our expectations, Grin2b$^{+/C456Y}$ mice showed normal social approach, social novelty recognition, and social interaction in three-chamber and direct social-interaction tests. In addition, these animals showed normal social communication (USVs) during courtship. Juvenile Grin2b$^{+/C456Y}$ mice also displayed normal social interaction in the juvenile play test and spent normal amounts of time with reunited mothers. It is possible, however, that the social tests and variables that we employed in the present study may not be sensitive enough to detect certain social deficits.

Both adult and juvenile Grin2b$^{+/C456Y}$ mice showed hypoactivity in the open-field test, suggesting that this phenotype is established early (in juvenile or earlier stages) and persist into adulthood. Adult Grin2b$^{+/C456Y}$ mice show anxiolytic-like behavior in the elevated plus-maze test but normal anxiety-like behaviors in open-field and light-dark tests. Juvenile Grin2b$^{+/C456Y}$ mice show anxiolytic-like behavior in the open-field test, suggesting that adult and juvenile Grin2b$^{+/C456Y}$ mice show normal or anxiolytic-like behaviors. In contrast, Grin2b$^{+/C456Y}$ pups display strongly increased USV calls upon mother separation, suggestive of anxiety-like behavior. Therefore, the anxiety-like behavior of Grin2b$^{+/C456Y}$ pups seems to be rapidly

weakened as these mice grow up, whereas self-grooming slowly develops at an adult stage, pointing to the contrasting trajectories of two important ASD-related phenotypes (anxiety-like behavior and self-grooming). How the early anxiety-like behavior in *Grin2b*$^{+/C456Y}$ pups is weakened or reversed as the pups grow into juveniles and adults remain unclear. This age-dependent reversal might reflect compensatory changes trying to overcome the over-activation of anxiety-related neural circuits. Although further details remain to be determined, our results are in line with the fact that anxiety is one of the key comorbidities of ASD [46,78] and that many mouse models of ASD display anxiety-like behaviors [42,79,80]. Notably, anxiolytic-like behavior has also been observed in mice lacking oxytocin [81], implicated in ASD [82].

*Grin2b*$^{+/−}$ mice (carrying conventional heterozygous *Grin2b* deletion) mice showed largely similar behaviors compared with those observed in *Grin2b*$^{+/C456Y}$ mice. Similar behaviors include hypoactivity in the open-field test and normal anxiety-like behavior in open-field and light-dark tests, but anxiolytic-like behavior of *Grin2b*$^{+/−}$ mice in the elevated plus-maze was much weaker than that in *Grin2b*$^{+/C456Y}$ mice. Biochemically, *Grin2b*$^{+/−}$ mice showed decreased levels of GluN2B but not GluN1 at P14 and P21, unlike the concomitant decreases in GluN2B and GluN1 in *Grin2b*$^{+/C456Y}$ mice, which may lead to subtle differences in synaptic and behavioral dysfunctions in these two mouse lines.

The behavioral phenotypes of *Grin2b*$^{+/C456Y}$ mice could not be compared with the symptoms of the human individual carrying GluN2B-C456Y mutation as they were minimally described in the previous study other than the fact that the mutation is a de novo mutation from a male individual with autism and intellectual disability [12]. However, the abnormal behaviors (i.e., anxiolytic-like behavior) of the mutant mice are important biologically because spending more time in the center region of a novel open-field arena or in the open arms of the elevated plus-maze reflects behaviors that would pose a significant threat for the survival of a mouse in its natural environment, implicating substantial deficits in cognitive functions. The anxiolytic-like behavior may not reflect increased fear of a dark or closed environment, because these mice exhibited a normal preference for light and dark chambers in the light-dark test. In addition, the anxiolytic-like behavior of the mutant mice in the elevated plus-maze does not seem to involve suppressed cognition of the fact that a darker and closed place is generally safe at least based on their normal learning and memory in Morris water-maze and novel object–recognition tests.

## NMDAR function, LTD, and anxiolytic-like behavior

Importantly, our study suggests synaptic mechanisms that may be associated with the anxiolytic-like behavior, namely suppressed NMDAR functions and LFS-LTD at an early stage. In support of this hypothesis, early chronic D-cycloserine treatment of young *Grin2b*$^{+/C456Y}$ mice normalizes NMDAR function and LTD in juvenile *Grin2b*$^{+/C456Y}$ mice and anxiolytic-like behavior in adult *Grin2b*$^{+/C456Y}$ mice. In addition, late acute treatment of adult *Grin2b*$^{+/C456Y}$ mice with D-cycloserine has no effect on abnormal behaviors probably because GluN2B expression is decreased at adult stages, and NMDAR-dependent LTD is difficult to induce at adult stages likely due to the switch of GluN2B to GluN2A [5,83,84]. The effect of late chronic D-cycloserine treatment in adult *Grin2b*$^{+/C456Y}$ mice could not be tested because the chronic treatment procedure seemed to increase anxiety levels in these mice, blunting the baseline difference between WT and mutant mice. In addition, we could not test whether the increased USV calls in the mutant pups are associated with the reduced NMDAR function because the time window for early D-cycloserine treatment (P7–16) fell behind that for pup USV testing (P4–12). Together, these results suggest that early correction of NMDAR function and NMDAR-dependent LTD in young mice leads to long-lasting improvement of anxiolytic-like

behavior in adult mice. Early treatment seems to be particularly important, not only because it has long-lasting effects, eliminating the necessity of repeated drug administration, but also because the small time window during which treatment is efficient appears to occur only during early developmental stages. Indeed, LTD is known to be most prominent during an early period (approximately 2–3 weeks) of postnatal brain development in mice and becomes weaker as the brain progressively matures and the ratio of GluN2B/GluN2A expression decreases [34,85–87].

Our findings are in line with the emerging concept that early and timely correction of key pathophysiological deficits in young mice is critical for the long-lasting and efficient rescue of synaptic and behavioral phenotypes in adult mice. For instance, early chronic fluoxetine treatment to restore reduced serotonin levels in young mice carrying a 15q11-13 duplication, a human ASD-risk mutation, has been shown to induce long-lasting normalization of serotonin levels and abnormal behaviors in adult mice [88]. Similarly, *Shank2*-mutant mice show increased NMDAR function (in contrast to decreased NMDAR function in later stages) [9], and early, but not late, chronic memantine treatment to suppress the abnormal NMDAR hyperfunction improves late synaptic and social phenotypes in *Shank2*-mutant mice [89].

Our data indicate that LFS-LTD are similarly decreased in the hippocampus and mPFC. These results suggest that the decreased NMDAR function and LFS-LTD in the hippocampus may represent a proxy for changes occurring in other brain regions and that decreased NMDAR function and LFS-LTD in many brain areas, additional to the hippocampus, could contribute to the behavioral changes (i.e., anxiolytic-like behavior) observed in *Grin2b*[+/C456Y] mice. In line with this idea, *Grin2b* is widely expressed in the brain [5], and anxiety-like behavior has been associated with various brain regions, including the hippocampus, anterior cingulate cortex, lateral septum, bed nuclei of the stria terminalis, paraventricular nucleus, and basolateral amygdala [90–97].

Lastly, GluN2B-C456Y is a strong ASD-risk mutation [12]. How might the abnormal synaptic and behavioral phenotypes of *Grin2b*[+/C456Y] mice be related to ASD pathophysiology? GluN2B-containing NMDARs that mediate large calcium influx are strongly expressed during early brain developmental stages to promote synapse and neuronal maturation through mechanisms, including posttranslational modification and gene expression [5]. Therefore, the decreased levels of GluN2B and GluN2B-containing NMDARs in *Grin2b*[+/C456Y] mice would suppress these critical molecular and cellular early events. In addition, NMDAR-dependent LTD during early brain development is well known to sharpen neuronal circuits by promoting weakening of less active synapses and strengthening of more active synapses through redistribution of synaptic protein resources between these synapses [83,84]. Therefore, reduced LTD in the developing brain of young *Grin2b*[+/C456Y] mice would suppress LTD-dependent synapse-pruning and circuit-sharpening processes, leading to brain malfunctions and abnormal behaviors. This prediction, based on in vivo results, might apply not only to GluN2B-C456Y-related cases of ASD [1,11–16] but also to various GRIN2B-related brain dysfunctions, including developmental delay, intellectual disability, attention-deficit/hyperactivity disorder, epilepsy, schizophrenia, obsessive-compulsive disorder, and encephalopathy [15,17]. How these predicted changes manifest into synaptic and circuit properties in the mutant brain remains to be determined. Previous studies have shown that NMDAR antagonists, including the GluN2B-specific antagonist ifenprodil, can induce anxiolytic-like behaviors in both humans and experimental animals [98]. However, a decrease in NMDAR function in the adult mutant brain is an unlikely possibility because both acute and chronic D-cycloserine treatment failed to rescue the anxiolytic-like behavior in adult *Grin2b*[+/C456Y] mice.

In conclusion, the heterozygous ASD-risk mutation, GluN2B-C456Y, leads to decreased GluN2B protein levels, diminished currents of GluN2B-containing NMDARs, and reduced

NMDAR-dependent LTD in young mice, as well as abnormal, anxiolytic-like behavior in adult mice. In addition, early D-cycloserine treatment of young mutant mice correcting NMDAR function and NMDAR-dependent LTD leads to long-lasting improvement of anxiolytic-like behaviors in adult mice.

## Materials and methods

### Ethics statement

All animals were bred and maintained according to the Requirements of Animal Research at KAIST and all procedures were approved by the Committees of Animal Research at KAIST (KA2016-31).

### Animals

Grin2b knock-in mice under the genetic background of C57BL/6J carrying C456Y mutation in exon 6 with Frt sites and cassette were designed and generated by Biocytogen (*Grin2b*$^{+/cassette}$, S1A Fig). To remove the neomycin cassette, *Grin2b*$^{+/cassette}$ mice were crossed with Protamine-Flp mice (C57BL/6J), which yielded floxed heterozygote mice (*Grin2b*$^{+/C456Y}$).

### Statistical analysis

Statistical analyses were performed using Prism GraphPad 7 and SigmaPlot 11. The data with nonparametric distribution were analyzed by Mann-Whitney test, and those with parametric distribution were analyzed by Student *t* test. If the data are parametric but have significant difference in variance in the F-test, Welch's correction was used. Including gender, age, and number of mice, all the details of the statistical analyses are described in S1 Data.

### Additional materials and methods

Details on other methods, including those for experiments on recombinant GluN1/GluN2B receptors, can be found in S2 Data.

### Numerical data

The numerical data used in all figures can be found in S3 Data.

### Original images for blots and gels

The original images for blots and gels can be found in S4 Data.

## Supporting information

**S1 Fig. Structural modeling predicts that the GluN2B-C456Y mutation disrupts a disulfide bond between the ATD and LBD.** (A and B) Molecular modeling of the GluN2B-C456Y protein in complex with the GluN1 subunit of NMDARs. Note that the two cysteine residues in patch 1 of the LBD and patch 2 of the ATD in the WT GluN2B protein form a disulfide bond that strengthens the interaction between LBD and ATD, a bond that is disrupted by the GluN2B-C456Y mutation in patch 1 of the LBD. ATD, amino-terminal domain; LBD, ligand-binding domain; NMDAR, N-methyl-D-aspartate receptor; WT, wild type.
(TIF)

**S2 Fig. The GluN2B-C456Y mutation decreases recombinant NMDAR currents and alters receptor properties.** (A) The GluN2B-C456Y mutation strongly decreases diheteromeric GluN1/GluN2B NMDAR currents in *Xenopus* oocytes. Note that the amount of the mutant

currents is <1% of the WT currents, despite the fact that mutant currents were recorded 1 day later than WT (3 and 2 days following oocyte injection, respectively). $n$ = 73 oocytes for WT (5.70 ± 0.61 μA) and 59 oocytes for C456Y (0.039 ± 0.004 μA), ***$P$ < 0.001, Mann-Whitney. (B) The GluN2B-C456Y mutation increases maximal open probability, as assessed by measuring MK-801 inhibition kinetics. $n$ = 22 oocytes for WT (1 ± 0.03, relative $\tau_{on}$) and 19 oocytes for C456Y (0.71 ± 0.05, relative $\tau_{on}$), ***$P$ < 0.001, Mann-Whitney. (C) The GluN2B-C456Y mutation reduces the sensitivity to extracellular protons. $n$ = 4 oocytes for WT (pH $IC_{50}$ = 7.49 ± 0.016) and 5 oocytes for C456Y (pH $IC_{50}$ = 7.11 ± 0.0075), *$P$ = 0.016, Mann-Whitney. (D) The GluN2B-C456Y mutation decreases the spermine-dependent potentiation. $n$ = 5 oocytes for WT (9.45 ± 0.51, spermine potentiation) and 4 oocytes for C456Y (2.83 ± 0.077, spermine potentiation),*$P$ = 0.016, Mann-Whitney. (E) The GluN2B-C456Y mutation does not affect the sensitivity to glutamate. $n$ = 4 oocytes for WT ($EC_{50}$ = 1.75 ± 0.04 μM) and 3 oocytes for C456Y ($EC_{50}$ = 1.86 ± 0.02 μM), $P$ = 0.23, Mann-Whitney. (F) The GluN2B-C456Y mutation decreases the sensitivity to glycine. $n$ = 4 oocytes for WT ($EC_{50}$ = 0.38 ± 0.017 μM) and 9 oocytes for C456Y ($EC_{50}$ = 1.13 ± 0.049 μM), **$P$ = 0.007, Mann-Whitney. (G) The GluN2B-C456Y mutation has minimal effect on the sensitivity to extracellular zinc. $n$ = 11 oocytes for WT ($IC_{50}$ = 0.68 ± 0.07 μM) and 11 oocytes for C456Y ($IC_{50}$ = 0.97 ± 0.1 μM), ***$P$ < 0.001, Mann-Whitney. (H) D-cycloserine is a partial agonist at GluN2B-C456Y mutant receptors. Currents recorded in 100 μM glutamate plus 100 μM D-cycloserine were normalized to currents recorded in 100 μM glutamate + 100 μM glycine (no D-cycloserine). $n$ = 9 oocytes for WT (relative current: 0.57 ± 0.005) and 9 oocytes for C456Y (relative current: 0.40 ± 0.006), ***$P$ < 0.001, Mann-Whitney. The numerical data underlying this figure can be found in S3 Data. $EC_{50}$, half maximal effective concentration; $IC_{50}$, half maximal inhibitory concentration; NMDAR, N-methyl-D-aspartate receptor; ns, not significant; WT, wild type. (TIF)

**S3 Fig. Knock-in strategy and PCR genotyping for the GluN2B-C456Y mutation in mice.** (A) Knock-in strategy for the GluN2B-C456Y mutation in mice. WT exon 6 was replaced with a mutant exon 6 containing the C456Y mutation. (B) PCR genotyping of homozygous ("Homo") and HT KI mice. Ex, exon; Frt, flippase target site; Homo, homozygous; HT, heterozygous; KI, knock-in; Neo, neomycin gene; WT, wild type. (TIF)

**S4 Fig. Decreased GluN2B and GluN1 protein levels, but normal *Grin2b* and *Grin1* mRNA levels, in *Grin2b*$^{+/C456Y}$ mice.** (A) Crude synaptosomal fractions from the *Grin2b*$^{+/C456Y}$ brain at multiple developmental stages (E20, P14, P21, P28, and P56) were immunoblotted with the indicated antibodies. For quantification (bar graphs), average levels of GluN1, Glu2A, and Glu2B proteins from *Grin2b*$^{+/C456Y}$ mice were normalized to those from WT mice. $n$ = 4 mice for WT and HT, *$P$ < 0.05, **$P$ < 0.01, ***$P$ < 0.001, Student $t$ test. (B) Normal levels of *Grin2b* and *Grin1* (encoding GluN1) mRNAs in WT, HT, and homozygous ("Homo") KI embryos (E20), as indicated by the results of RT-qPCR reactions targeting Grin2b mRNA exons 3, 4, 11, or 14, and Grin1 mRNA exons 3, 7, or 12. $n$ = 4 mice for WT, 4 for HT, and 3 for Homo, one-way ANOVA with Tukey's test. The numerical data underlying this figure can be found in S3 Data. E, embryonic day; HT, heterozygous; KI, knock-in; ns, not significant; P, postnatal day; RT-qPCR, real-time quantitative PCR; WT, wild type. (TIF)

**S5 Fig. Spontaneous and evoked synaptic transmission at excitatory and inhibitory synapses, as well as neuronal excitability, are normal in *Grin2b*$^{+/C456Y}$ hippocampal CA1 neurons.** (A) Normal mEPSCs in CA1 neurons of *Grin2b*$^{+/C456Y}$ mice (P18–20). $n$ = 15 neurons

from 3 mice for WT and 15 (3) for HT, Mann-Whitney test (frequency) and Student *t* test (amplitude). (B) Normal mIPSCs in CA1 neurons of *Grin2b*$^{+/C456Y}$ mice (P21–23). *n* = 15 (3) for WT and HT, Student *t* test. (C) Normal sEPSCs in CA1 neurons of *Grin2b*$^{+/C456Y}$ mice (P22–24). *n* = 15 (3) for WT and 14 (4) for HT, Mann-Whitney test. (D) Normal sIPSCs in CA1 neurons of *Grin2b*$^{+/C456Y}$ mice (P22–24). *n* = 13 (3) for WT and 18 (4) for HT, Mann-Whitney test (frequency) and Student *t* test (amplitude). (E) Normal ratio of evoked IPSCs over evoked EPSCs in the CA1 region of *Grin2b*$^{+/C456Y}$ mice (P20–22). *n* = 8 (4) for WT and 8 (3) for HT, Mann-Whitney test. (F) Normal neuronal excitability in CA1 neurons of *Grin2b*$^{+/C456Y}$ mice (P21–23), as indicated by the current-firing relationship. *n* = 13 (3) for WT and 13 (4) for HT, two-way ANOVA. The numerical data underlying this figure can be found in S3 Data. EPSC, excitatory postsynaptic current; HT, heterozygous; IPSC, inhibitory postsynaptic current; mEPSC, miniature EPSC; mIPSC, miniature IPSC; ns, not significant; P, postnatal day; sEPSC, spontaneous EPSC; sIPSC, spontaneous IPSC; WT, wild type.
(TIF)

**S6 Fig. Normal spatial and recognition learning and memory in *Grin2b*$^{+/C456Y}$ mice.** (A–E) Normal spatial learning and memory in *Grin2b*$^{+/C456Y}$ mice (P90–114) in the learning (A) and probe (B–E) phases of the initial (A–C) and reversal (A, D, E) sessions of the Morris water maze, as shown by time taken to escape to the platform, percent of time spent in target quadrant, and number of crossing over the former platform location. *n* = 12 mice for WT and HT, two-way ANOVA with Sidak's test and Student *t* test. (F) Normal novel object–recognition memory in *Grin2b*$^{+/C456Y}$ mice (P70–80), as shown by the percent of time spent exploring a novel object relative to a familiar object. *n* = 12 mice for WT and HT, Student *t* test. The numerical data underlying this figure can be found in S3 Data. HT, heterozygous; ns, not significant; P, postnatal day; WT, wild type.
(TIF)

**S7 Fig. Normal social interaction and communication and moderately enhanced self-grooming in *Grin2b*$^{+/C456Y}$ mice.** (A–H) Normal social approach and social novelty recognition in *Grin2b*$^{+/C456Y}$ mice (P79–80) in the three-chamber test under both bright-light and dark conditions, as shown by time spent sniffing the target and time spent in the chamber with the target. *n* = 8 mice for WT and HT (except for *n* = 7 for HT for social novelty recognition), $^{**}P < 0.01$, $^{***}P < 0.001$, two-way ANOVA with Sidak's test. (I–K) Normal social interaction in freely moving *Grin2b*$^{+/C456Y}$ mice (P62–89) in the direct social-interaction test, as shown by time spent in nose-to-nose sniffing or following and total time spent in social interaction (sniffing, following, and other social interactions). *n* = 11 mice for WT and 13 for HT, Student *t* test (except for Mann-Whitney test for following). (L–N) Normal courtship USVs in male *Grin2b*$^{+/C456Y}$ mice (P74–101) upon encountering a novel female mouse, as shown by the number of calls, duration of each call, and latency to the first call. *n* = 21 mice for WT and 17 for HT, Student *t* test (except for Mann-Whitney test for latency to first call). (O–P) Enhanced repetitive self-grooming (but normal digging) by *Grin2b*$^{+/C456Y}$ mice (P68–88) in home cages with bedding, but normal self-grooming in a novel chamber without bedding, as shown by time spent self-grooming (or digging). *n* = 21 mice for WT and 17 for HT for self-grooming in home cages with bedding, *n* = 16 mice for WT and HT for self-grooming in a novel chamber without bedding, $^{*}P < 0.05$, Student *t* test. The numerical data underlying this figure can be found in S3 Data. HT, heterozygous; ns, not significant; P, postnatal day; USV, ultrasonic vocalization; WT, wild type.
(TIF)

**S8 Fig. *Grin2b*$^{+/−}$ mice display decreased GluN2B levels and hypoactivity and anxiolytic-like behavior, similar to *Grin2b*$^{+/C456Y}$ mice.** (A and B) Decreased levels of the GluN2B, but

not GluN1 or GluN2A, protein in the *Grin2b*$^{+/-}$ mice brain. Whole-brain total lysates from *Grin2b*$^{+/-}$ mice at P14 and P21 were immunoblotted with anti-GluN1/2A/2B antibodies. For quantification, average levels of GluN1/2A/2B proteins from *Grin2b*$^{+/-}$ mice were normalized to those from WT mice. $n = 4$ mice for WT and HT, $^*P < 0.05$, $^{**}P < 0.01$, Student *t* test. (C–F) Hypoactivity and normal anxiety-like behavior in *Grin2b*$^{+/-}$ mice (P61–71) in the open-field test, as shown by distance moved and time spent in the center region of the open-field arena. $n = 16$ mice for WT and 14 for HT, $^*P < 0.05$, $^{**}P < 0.01$, $^{***}P < 0.001$, two-way ANOVA with Sidak's test and Student *t* test. (G–J) Anxiolytic-like behavior in *Grin2b*$^{+/-}$ mice (P65–75) in the elevated plus-maze, as shown by the number of entries into and time spent in open/ closed arms. $n = 16$ mice (WT) and 14 (HT), $^{***}P < 0.001$, Student *t* test. (K) Normal anxiety-like behavior in *Grin2b*$^{+/-}$ mice (P68–78) in the light-dark test, as shown by time spent in the light chamber. $n = 16$ mice (WT) and 14 (HT), Mann-Whitney test. The numerical data underlying this figure can be found in S3 Data. HT, heterozygous; ns, not significant; P, post-natal day; WT, wild type.
(TIF)

**S9 Fig. Normal social interaction and communication, repetitive behavior, and object memory in *Grin2b*$^{+/-}$ mice.** (A–H) Normal social approach and social novelty recognition in *Grin2b*$^{+/-}$ mice (P75–85) in the three-chamber test under both bright-light and dark conditions, as shown by time spent sniffing the target and time spent in the chamber with the target. $n = 8$ mice for WT and 7 for HT, $^*P < 0.05$, $^{**}P < 0.01$, $^{***}P < 0.001$, two-way ANOVA with Sidak's test. (I–K) Normal courtship USVs in *Grin2b*$^{+/-}$ mice (P77–87) upon encountering a novel female mouse, as shown by the number of calls, duration of each call, and latency to the first call. $n = 16$ for WT and 14 for HT, Student *t* test except Mann-Whitney test for latency to first all. (L and M) Normal repetitive self-grooming and digging in *Grin2b*$^{+/-}$ mice (P77–87) in home cages with bedding, as shown by time spent self-grooming or digging. $n = 16$ (WT) and 14 (HT), Mann-Whitney test. (N) Normal novel object–recognition memory in *Grin2b*$^{+/-}$ mice (P63–73), as shown by the percent of time spent exploring a novel object relative to a familiar object. $n = 16$ (WT) and 14 (HT), Student *t* test. The numerical data underlying this figure can be found in S3 Data. HT, heterozygous; ns, not significant; P, postnatal day; USV, ultrasonic vocalization; WT, wild type.
(TIF)

**S10 Fig. Normal social interaction and repetitive behavior in *Grin2b*$^{+/C456Y}$ juveniles.** (A and B) *Grin2b*$^{+/C456Y}$ juveniles (P22–24) show normal behaviors in the maternal-homing test, as shown by the time spent with the reunited mother. Note that these mice showed normal exploration of the bedding materials from previous home cages ("Home") suggestive of normal olfactory function; it is unclear why these mice prefer to explore the new corner ("New"). $n = 24$ mice for WT and 20 for HT, $^*P < 0.05$, $^{***}P < 0.001$, two-way ANOVA with Sidak's test. (C) *Grin2b*$^{+/C456Y}$ juveniles (P28–20) show normal social interaction in the juvenile play test, as shown by the total time spent in social interaction (nose-to-nose sniffing, following, and other social interactions). $n = 9$ pairs for WT and 5 pairs for HT, Student *t* test. (D and E) *Grin2b*$^{+/C456Y}$ juveniles (P24–26) show normal repetitive self-grooming and digging in home cages with bedding, as shown by time spent self-grooming or digging. $n = 24$ mice for WT and 20 for HT, Mann-Whitney test. The numerical data underlying this figure can be found in S3 Data. HT, heterozygous; ns, not significant; P, postnatal day; WT, wild type.
(TIF)

**S11 Fig. Early DCS rescues LFS-LTD and has no effect on the hypoactivity or anxiety-like behavior in *Grin2b*$^{+/-}$ mice.** (A) Experimental strategy for chronic, oral DCS treatment (40

mg/kg), twice daily for 10 days (P7–16), in young *Grin2b*$^{+/−}$ mice followed by LTD measurements in juvenile mice (P17–21) and behavioral tests (OFT and EPM) in adult mice (>P56). (B) Early chronic DCS treatment normalizes LFS-LTD at SC-CA1 synapses in juvenile *Grin2b*$^{+/−}$ mice (P17–20). *n* = 14 mice (6) for WT_V (78.0% ± 4.1%), 15 (5) for WT_D (85.2% ± 4.2%), 17 (6) for HT_V (96.9% ± 4.5%), 14 (5) for HT_D (79.6% ± 3.7%), *$P < 0.05$, two-way ANOVA with Tukey's test. (C) Early chronic DCS treatment has no effect on the hypoactivity in adult *Grin2b*$^{+/−}$ mice (P56–62). *n* = 11 mice for WT_D, 12 for WT_D, 9 for HT_V, 11 for HT_D, ***$P < 0.001$, two-way ANOVA with Tukey's test. (D) Early chronic DCS treatment has no effect on the anxiety-like behavior in adult *Grin2b*$^{+/−}$ mice (P58–76), as shown by closed-arm time and open-arm entry/time. Note, however, that the early chronic drug treatment procedure (P7–16; twice a day for 10 days; oral) seems to modestly increase anxiety levels in the mutant mice, blunting the baseline difference in the closed-arm entry between WT-V and HT-V (see also panel G in S8 Fig), making it impossible to assess the effect of DCS on this value. *n* = 11 mice for WT_D, 12 for WT_D, 9 for HT_V, 11 for HT_D, two-way ANOVA with Tukey's test. The numerical data underlying this figure can be found in S3 Data. DCS, D-cycloserine; EPM, elevated plus-maze; HT, heterozygous; HT_D, heterozygous with DCS; HT_V, heterozygous with vehicle; LFS, low-frequency stimulation; LTD, long-term depression; ns, not significant; OFT, open-field test; P, postnatal day; SC-CA1, Schaffer collateral-CA1 pyramidal; WT, wild type; WT_D, WT with DCS; WT_V, WT with vehicle.
(TIF)

**S12 Fig. Late chronic DCS treatment has no effect on anxiolytic-like behavior or hypoactivity in adult *Grin2b*$^{+/C456Y}$ mice.** (A–E) Late chronic oral DCS treatment (40 mg/kg; twice daily for 10 days) in adult *Grin2b*$^{+/C456Y}$ mice (P57–66) does not affect anxiolytic-like behavior in the EPM. Note, however, that the chronic drug treatment procedure (oral drug administration twice a day for 10 days using a restrainer, unlike the pup situation in which pups were gently grabbed) seems to substantially increase anxiety levels in the mutant mice, blunting the baseline difference in EPM variables between WT-V and HT-V (see also Fig 3E–3H and Fig 4F–4I), making it impossible to assess the effect of DCS on these values. *n* = 10 mice for WT_D, 11 for WT_D, 11 for HT_V, 12 for HT_D, two-way ANOVA with Tukey's test. (F–I) Late chronic oral DCS treatment (40 mg/kg; twice daily for 10 days) in adult *Grin2b*$^{+/C456Y}$ mice (P57–66) does not affect hypoactivity in the OFT. *n* = 10 mice for WT_D, 11 for WT_D, 11 for HT_V, 12 for HT_D, **$P < 0.01$, ***$P < 0.001$, two-way ANOVA with Tukey's test. The numerical data underlying this figure can be found in S3 Data. DCS, D-cycloserine; EPM, elevated plus-maze; HT, heterozygous; HT_D, heterozygous with DCS; HT_V, heterozygous with vehicle; ns, not significant; OFT, open-field test; P, postnatal day; WT, wild type; WT_D, WT with DCS; WT_V, WT with vehicle.
(TIF)

**S1 Data. Age and sex of mice used in the study and statistical methods and results.**
(XLSX)

**S2 Data. Supplementary methods.**
(DOCX)

**S3 Data. Numerical data underlying main figures (Figs 1–5) and supplementary figures (S2 and S4–S12 Fig).**
(XLSX)

**S4 Data. Uncropped immunoblot and gels images for Fig 1B, S3B Fig, S4A Fig, S8A Fig and S8B Fig.**
(PDF)

## Acknowledgments

We would like to thank Dr. Yeonseung Chung in the Department of Mathematical Sciences at KAIST for help with the statistical analyses.

## Author Contributions

**Conceptualization:** Wangyong Shin, Eun-Jae Lee, Eunjoon Kim.

**Data curation:** Wangyong Shin, Eun-Jae Lee, Hyejin Lee.

**Formal analysis:** Wangyong Shin, Kyungdeok Kim, Doyoun Kim, Muwon Kang, Eun-Jae Lee, Hyejin Lee.

**Funding acquisition:** Eunjoon Kim.

**Investigation:** Wangyong Shin, Kyungdeok Kim, Benjamin Serraz, Yi Sul Cho, Muwon Kang, Eun-Jae Lee, Hyejin Lee, Eunjoon Kim.

**Methodology:** Wangyong Shin, Kyungdeok Kim, Benjamin Serraz, Yi Sul Cho, Eun-Jae Lee, Hyejin Lee.

**Project administration:** Yong Chul Bae, Pierre Paoletti, Eunjoon Kim.

**Software:** Doyoun Kim.

**Supervision:** Yong Chul Bae, Pierre Paoletti, Eunjoon Kim.

**Validation:** Wangyong Shin.

**Visualization:** Wangyong Shin, Doyoun Kim.

**Writing – original draft:** Wangyong Shin, Pierre Paoletti, Eunjoon Kim.

**Writing – review & editing:** Wangyong Shin, Yong Chul Bae, Pierre Paoletti, Eunjoon Kim.

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
