## [Editor Report · Decision Letter 0]

27 Feb 2020

Dear Eunjoon, 

Thank you for submitting your revised manuscript entitled "Early correction of synaptic long-term depression improves abnormal anxiety-like behavior in adult GluN2B-C456Y-mutant mice" for consideration as a Research Article by PLOS Biology.

Your revision has now been evaluated by the PLOS Biology editorial staff, as well as by the original Academic Editor, and I am writing to let you know that we would like to send your submission out for external peer review.

Please re-submit your manuscript within two working days, i.e. by Mar 02 2020 11:59PM.

Kind regards,

Gabriel Gasque, Ph.D.,

Senior Editor

PLOS Biology

---

## [Decision Letter · Decision Letter 1]

26 Mar 2020

Dear Eunjoon,

Thank you for submitting your revised Research Article entitled "Early correction of synaptic long-term depression improves abnormal anxiety-like behavior in adult GluN2B-C456Y-mutant mice" for publication in PLOS Biology. I have now obtained advice from the original reviewers and have discussed their comments with the Academic Editor. 

Based on the reviews, we will probably accept this manuscript for publication, assuming that you will modify the manuscript to address the remaining points raised by reviewer 2. Please also make sure to address the data and other policy-related requests noted at the end of this email.

We expect to receive your revised manuscript within two weeks. Your revisions should address the specific points made by reviewer 2. Please submit the following files along with your revised manuscript:

In addition to the remaining revisions and before we will be able to formally accept your manuscript and consider it "in press", we also need to ensure that your article conforms to our guidelines. A member of our team will be in touch shortly with a set of requests. As we can't proceed until these requirements are met, your swift response will help prevent delays to publication.

*Copyediting*

*Published Peer Review History*

*Early Version*

*Submitting Your Revision*

Sincerely,

Gabriel Gasque, Ph.D., 

Senior Editor

PLOS Biology

DATA POLICY:

--Please update your S3 Data file to include Figure S2C.

Reviewer remarks:

Reviewer #1: The authors have adequately addressed the majority of this reviewer's concerns. With the significant changes made to the manuscript during the revision, the paper is suitable for publication.

Reviewer #2, Marc Fuccillo: The authors have attempted to address many of my requests. In doing so, they have strengthened the idea that the C456Y point mutant functions like the NR2B heterozygous LOF mutation, both at the synaptic, protein and behavioral level. While perhaps expected, these data provide clarity to the role of ASD-associated NMDAR mutations. Another strong plus is the addition of data showing a similar plasticity effect in the mPFC. these data suggest that the data in hippo are a proxy for other areas (something that should be mentioned in the text). 

Overall, I feel like this is a strong contribution to the asd pathophysiology literature. while not incredibly surprising, the data is clear and makes a strong point supporting this specific disease-associated mutation as GRIN2B haploinsufficiency. the reproducibility of DCS rescue across assays is also a strength. 

A few short textual things should be added:

1. discuss the relationship between anxiogenic and anxiolytic behaviors as they relate to ASD. these are not part of the core behavioral phenotype but are the most clearly altered in this work. the literature on ASD and anxiety-related behaviors should be discussed.

2. please discuss that in light of the mPFC data, it is unclear which brain regions are contributing to the mutation-associated behavioral (and rescue-related) changes.

3. "suppressed recognition of the fact that a dark and closed place is generally safe" - this isn't any better than before - perhaps just describe the phenotype.

Reviewer #3: The authors performed extra experiments and gave explanations to address all the questions raised by the reviewers. The manuscript became even more data heavy and the conclusions are now better discussed to support the authors ideas. Although there are some technical issues, the authors do demonstrate GluN2B-C456Y haploinsufficiency decreases GluN2B protein levels, LTD, and anxiety-like behavior. The overall behavioral and physiological experiments provide an insight for the importance of early correction of pathophysiological deficits.

The findings are important in the field for the treatment of neurodevelopmental or psychiatric diseases caused by NMDAR mutations.

Reviewer #4: Satisfied with the manuscript as edited. Again, it is an important contribution to the literature as I stated in my initial review, though there are not major new insights here. Addition of the PFC data at least provides some additional evidence for more general deficits.

---

## [Editor Report · Decision Letter 2]

15 Apr 2020

Dear Dr Kim,

On behalf of my colleagues and the Academic Editor, Thomas C Südhof, I am pleased to inform you that we will be delighted to publish your Research Article in PLOS Biology. 

Early Version

PRESS 

Kind regards,

Vita Usova

Publication Editor, 

PLOS Biology

on behalf of

Gabriel Gasque,

Senior Editor

PLOS Biology